# A Light-Powered Liquid Crystal Elastomer Spring Oscillator with Self-Shading Coatings

**DOI:** 10.3390/polym14081525

**Published:** 2022-04-09

**Authors:** Changshen Du, Quanbao Cheng, Kai Li, Yong Yu

**Affiliations:** Department of Civil Engineering, Anhui Jianzhu University, Hefei 230601, China; changshendu@yeah.net (C.D.); cheng_quanbao@outlook.com (Q.C.); kli@ahjzu.edu.cn (K.L.)

**Keywords:** spring oscillator, light-powered mechanisms, liquid crystal elastomer, self-shading effect, limit cycles

## Abstract

The self-oscillating systems based on stimuli-responsive materials, without complex controllers and additional batteries, have great application prospects in the fields of intelligent machines, soft robotics, and light-powered motors. Recently, the periodic oscillation of an LCE fiber with a mass block under periodic illumination was reported. This system requires periodic illumination, which limits the application of self-sustained systems. In this paper, we creatively proposed a light-powered liquid crystal elastomer (LCE) spring oscillator with self-shading coatings, which can self-oscillate continuously under steady illumination. On the basis of the well-established dynamic LCE model, the governing equation of the LCE spring oscillator is formulated, and the self-excited oscillation is studied theoretically. The numerical calculations show that the LCE spring oscillator has two motion modes, static mode and oscillation mode, and the self-oscillation arises from the coupling between the light-driven deformation and its movement. Furthermore, the contraction coefficient, damping coefficient, painting stretch, light intensity, spring constant, and gravitational acceleration all affect the self-excited oscillation of the spring oscillator, and each parameter is a critical value for triggering self-excited oscillation. This work will provide effective help in designing new optically responsive structures for engineering applications.

## 1. Introduction

Self-excited oscillation is a kind of periodic motion maintained by constant external stimulation [1,2,3,4,5,6]. Similar to biological active feeding, it can directly harvest energy from a constant environment to maintain its periodic motion [7,8,9,10,11]. This feature means that the self-excited oscillation system has no requirement for complex controllers or heavy batteries, which makes it more stable and portable [12,13,14,15,16]. In addition, the period and amplitude of self-excited oscillation generally depend on the intrinsic parameters of the system and have nothing to do with the initial conditions, which makes the system robust [17,18,19,20,21]. Many self-excited oscillation systems, constructed using conventional or active materials, have been studied [22,23,24,25].

The stimuli-responsive materials of the self-excited oscillation system include hydrogels [26,27,28,29], ion gels [30,31], and liquid crystal elastomer (LCE) [32,33,34,35]. Using different stimuli-responsive materials and structures, different feedback mechanisms have been proposed to realize energy compensation, such as the coupling mechanism between a chemical reaction and large deformation [36], the self-shading mechanism [17,37], and a coupling mechanism in droplet evaporation multi-processes [38]. These mechanisms are originated from the nonlinear coupling of multiple processes for implementing feedback. Self-excited oscillation based on active materials has a wide application prospect in many fields, such as energy acquisition, sensing with electronic skins [39], energy harvesters [13,14,15], soft robotics [16,17,18,19], active machines [8,12], and motors [5,7].

LCE is a novel material among the stimuli-responsive polymer materials, and is a polymer network structure formed by cross-linking liquid crystal monomer molecules [40]. This special composition and structure enable LCE to respond to external excitations, such as heat [41,42,43], electric fields [44,45], light [33,35,46,47], magnetic fields [48], and chemicals [26,27,31]. LCE generally has the advantages of rapid deformation response, recoverable deformation, and silence [32,33,34,35]. In recent years, a great number of self-oscillating systems based on LCE materials have attracted widespread attention. Using LCE materials, several self-excited motion modes have been constructed, such as rolling [24], vibration [24,49,50], swinging [51], stretching and shrinking [46], rotation [25,32], eversion or inversion [43,52], torsion [53], jumping [54,55,56], buckling [35,57], and even the synchronization and group behavior of several coupled self-excited oscillators [58].

Recently, the periodic oscillation of an LCE fiber with a mass block in a periodic illumination was reported [10], and a schematic diagram of the oscillating system is shown in Figure 1a. The results of this experiment are represented in Figure 1b. The mechanism of this oscillation is simple: in an illuminated state, the LCE fiber contracts and pulls the mass block upwards. Then, in a non-illuminated state, the deformation of the LCE fiber resumes and the mass moves downwards because of gravity. Similarly, existing self-oscillating systems need illumination in special directions and locally distributed structured light, which imposes additional requirements on the system and limits the application of self-oscillating systems.

Inspired by the experiment as shown in Figure 1, we creatively proposed a spring oscillator based on a self-shading mechanism, which is composed of an optically responsive LCE fiber with self-shading coatings and a mass block, as shown in Figure 2. The system can self-oscillate when it is exposed to steady illumination, which has many advantages, such as wide applicability, simplicity, reliability, and ease of adjustment. In this paper, we will investigate the self-excited oscillation of the LCE spring oscillator under steady illumination based on the well-established dynamic LCE model, reveal the mechanism of the self-excited oscillation, and quantitatively analyze the influence of several key physical quantities on its motion modes, amplitude, and period. This work will provide effective help in designing new optically responsive machines for engineering applications.

## 2. Theoretical Model and Formulation

### 2.1. Dynamics of an LCE Spring Oscillator Coated with an Opaque Powder Coating

Figure 2 sketches a dynamic model of an LCE spring oscillator under steady illumination, which is composed of an optically responsive LCE fiber with self-shading coatings and a mass block. The shaded area in the figure represents the illuminated region. The original length of the LCE fiber in a stress-free state is denoted by $L_{0}$ (Figure 2a), and the photosensitive molecules, such as the azobenzene molecules in the nematic LCE fibers, are oriented along the fiber axis, as shown in Figure 2b. To obtain the light-powered LCE spring oscillator with self-shading coatings by fabricating LCE rods [32] made of liquid crystal monomers (RM257) and cross-linkers (PETMP), etc., using the two-step cross-linking reaction reported by Yakacki et al. [59], one can first fabricate a polydomain LCE fiber, and a mechanical stress is applied to orient the mesogens into a monodomain. The shading coating can be prepared with a shading powder, such as titanium dioxide. The LCE fiber is pre-stretched to the painting stretch $L_{p}$, and then one can use a small amount of glue to apply the shading coating on the surface of the LCE fiber, and one will obtain the light-powered LCE spring oscillator with self-shading coatings (Figure 2b). One end of the fiber is suspended on the fixed top, and the other end is tied with a mass block with mass m. During the oscillation, the position of the mass block is denoted by u(t), and the current length of the LCE fiber is $l\left(t\right)=L_{0}+u\left(t\right)$ (Figure 2c). Initially, the mass block is located at position $u_{0}$, and an initial velocity ${\dot{u}}_{0}$ is given. Under the action of gravity and the pulling force of the LCE fiber, the mass block starts to oscillate.

Define the dimensionless quantities $\tilde{l}\left(t\right)=l\left(t\right)/L_{0}$ and ${\tilde{L}}_{p}=L_{p}/L_{0}$, and the shading effects of the shading coating are shown in Figure 3. When $l\left(t\right)>L_{p}$, the shading coating will produce a large number of irregular cracks, so the light intensity on the surface of the LCE fiber increases rapidly. For simplicity, we assume that for $l\left(t\right)>L_{p}$, the LCE fiber is in an illuminated state, and the liquid crystal molecules in the LCE fiber change from trans to *cis*. For $l\left(t\right)\leq L_{p}$, the LCE fiber is in a non-illuminated state, and the liquid crystal molecules change from *cis* to trans, which diminishes the light-driven contraction of the LCE fiber. Therefore, the spring force in the LCE fiber may vary periodically, and eventually self-excited oscillation may be triggered under uniform and constant illumination conditions by the self-shading effect of the coating.

The mass block is subjected to the spring force F(t) of the LCE fiber, the damping force $F_{d}\left(t\right)$, and the gravitational force. For simplicity, it is assumed that the air damping force is proportional to the velocity of the mass, and its direction is always opposite to the velocity of the mass. The mass of the LCE fiber is assumed to be much smaller than that of the mass, and thus it is negligible. According to Newtonian mechanics, the following governing equation holds at any moment during the oscillation of the mass:(1)$$m\ddot{u}\left(t\right)=mg-F\left(t\right)-c\dot{u}\left(t\right),$$
where g is the gravitational acceleration, c is the damping coefficient, and $\dot{u}$ and $\ddot{u}$ indicate the velocity $\frac{du\left(t\right)}{dt}$ and acceleration $\frac{d^{2}u\left(t\right)}{dt^{2}}$ of the mass, respectively.

In Equation (1), the spring force of the LCE fiber is assumed to be linear to the strain, as per the following form:(2)$$F\left(t\right)=k\left[u\left(t\right)-L_{0}\varepsilon \left(t\right)\right],$$
where k is the spring constant of the LCE fiber, and the light-driven contraction strain ε(t) is assumed to be linear to the number fraction ϕ(t) of the *cis* number fraction in the LCE fiber, which can be written as
(3)ε(t)=−αϕ(t),
where α is the contraction coefficient and takes its absolute value for calculation.

### 2.2. Dynamics Model of the LCE Fiber

The experiments show that UV light with a wavelength of less than 400 nm or a laser can induce the trans to *cis* isomerization [60]. Generally, under UV light excitation, light-driven *cis* to trans isomerization can be neglected [61], and the number fraction of *cis*-isomers ϕ(t) depends on thermal excitation from trans to *cis*, thermally driven relaxation from *cis* to trans, and light-driven trans to *cis* isomerization. Furthermore, the thermal excitation from trans to *cis* is often negligible relative to the light-driven excitation [61]. In this paper, we use the following governing equation to describe the evolution of the number fraction ϕ(t) of the *cis* number fraction [62]:(4)$$\frac{\partial \phi \left(t\right)}{\partial t}=\eta_{0}I_{0}\left[1-\phi \left(t\right)\right]-T_{0}^{-1}\phi \left(t\right),$$
where $T_{0}$ is the thermal relaxation time from the *cis* state to the trans state, $I_{0}$ is the light intensity, and $\eta_{0}$ is a light absorption constant. The solution to Equation (4) can be easily obtained as
(5)$$\phi \left(t\right)=\frac{\eta_{0}T_{0}I_{0}}{\eta_{0}T_{0}I_{0}+1}+\left(\phi_{0}-\frac{\eta_{0}T_{0}I_{0}}{\eta_{0}T_{0}I_{0}+1}\right)\exp\left[-\frac{t}{T_{0}}\left(\eta_{0}T_{0}I_{0}+1\right)\right],$$
where *ϕ*_0_ is the number fraction of *cis*-isomers at *t* = 0. In the illuminated state, for initially zero number fractions of *cis*-isomers, i.e., *ϕ*_0_ = 0, Equation (5) can be simplified as
(6)$$\phi (t)=\frac{\eta_{0}T_{0}I_{0}}{\eta_{0}T_{0}I_{0}+1}\left\{1-\exp\left[-\frac{t}{T_{0}}\left(1+\eta_{0}T_{0}I_{0}\right)\right]\right\}.$$

In the non-illuminated state, namely, *I*_0_ = 0, Equation (5) can be simplified as
(7)$$\phi \left(t\right)=\phi_{0}\exp\left(-\frac{t}{T_{0}}\right).$$
where *ϕ*_0_ can be set as the maximum of *ϕ*(*t*) in Equation (6), namely, $\phi_{0}=\frac{\eta_{0}T_{0}I_{0}}{\eta_{0}T_{0}I_{0}+1}$, and Equation (1) can be simplified as
(8)$$\phi \left(t\right)=\frac{\eta_{0}T_{0}I_{0}}{\eta_{0}T_{0}I_{0}+1}\exp\left(-\frac{t}{T_{0}}\right).$$

By defining the dimensionless quantities $\tilde{I}=\eta_{0}T_{0}I_{0}$, $\tilde{t}=t/T_{0}$, and $\tilde{\phi }=\phi \left(\eta_{0}T_{0}I_{0}+1\right)/\eta_{0}T_{0}I_{0}$, in the illuminated state, Equation (6) is rewritten as
(9)$$\tilde{\phi }=1-\exp\left[-\tilde{t}\left(\tilde{I}+1\right)\right],$$
and in the non-illuminated state, Equation (8) is rewritten as
(10)$$\tilde{\phi }=\exp\left(-\tilde{t}\right).$$

### 2.3. Governing Equations of the LCE Spring Oscillator

We define the following dimensionless parameters: $\tilde{F}\left(t\right)=F\left(t\right)T_{0}^{2}/mL_{0}$, $\tilde{u}\left(t\right)=u\left(t\right)/L_{0}$, $\tilde{c}=cT_{0}/m$, $\tilde{g}=gT_{0}^{2}/L_{0}$, and $\tilde{k}=kT_{0}^{2}/m$. Thus, Equation (1) can be rewritten in a dimensionless form as
(11)$$\tilde{\ddot{u}}=\tilde{g}-\tilde{k}\left[\tilde{u}\left(\tilde{t}\right)+\alpha \phi \right]-\tilde{c}\tilde{\dot{u}},$$
where $\tilde{\dot{u}}$ and $\tilde{\ddot{u}}$ indicate the velocity $\frac{d\tilde{u}\left(\tilde{t}\right)}{d\tilde{t}}$ and acceleration $\frac{d^{2}\tilde{u}\left(\tilde{t}\right)}{dt^{2}}$ of the mass, respectively.

Combing Equations (9)–(11) leads to:

In an illuminated state, i.e., $\tilde{l}\left(t\right)>{\tilde{L}}_{p}$,
(12)$$\tilde{\ddot{u}}=\tilde{g}-\tilde{k}\left\{\tilde{u}\left(\tilde{t}\right)+\alpha \tilde{I}\frac{1-\exp\left[-\tilde{t}\left(\tilde{I}+1\right)\right]}{\tilde{I}+1}\right\}-c\tilde{\dot{u}},$$

In a non-illuminated state, i.e., $\tilde{l}\left(t\right)\leq {\tilde{L}}_{p}$,
(13)$$\tilde{\ddot{u}}=\tilde{g}-\tilde{k}\left[\tilde{u}\left(\tilde{t}\right)+\alpha \tilde{I}\frac{\exp\left(-\tilde{t}\right)}{\tilde{I}+1}\right]-c\tilde{\dot{u}}.$$

### 2.4. Solution Method

Equations (12) and (13) are ordinary differential equations with variable coefficients, and there exists no analytical solutions. Hereon, the classical fourth-order Runge–Kutta method is used to numerically solve the ordinary differential equations using the software MATLAB. We first transform the second-order ordinary differential equation into two first-order ordinary differential equations. Therefore, the governing equations are rewritten as
(14)$$\left\{ \begin{array}{l} \tilde{\dot{u}}=z\\ \dot{z}=f\left(\tilde{t},\tilde{u},z\right)\\ z\left(t_{0}\right)={\tilde{\dot{u}}}_{0}\\ \tilde{u}\left(t_{0}\right)={\tilde{u}}_{0} \end{array}\right..$$

The formula for the fourth-order Runge–Kutta is
(15)$$\left\{ \begin{array}{l} {\tilde{u}}_{n+1}={\tilde{u}}_{n}+hz_{n}+\frac{h^{2}}{6}\left(L_{1}+L_{2}+L_{3}\right)\\ z_{n+1}=z_{n}+\frac{h}{6}\left(L_{1}+2L_{2}+2L_{3}+L_{4}\right) \end{array}\right.,$$
(16)$$\left\{ \begin{array}{l} L_{1}=f\left({\tilde{t}}_{n},{\tilde{u}}_{n},z_{n}\right)\\ L_{2}=f\left({\tilde{t}}_{n}+\frac{h}{2},{\tilde{u}}_{n}+\frac{h}{2}z_{n},z_{n}+\frac{h}{2}L_{1}\right)\\ L_{3}=f\left({\tilde{t}}_{n}+\frac{h}{2},{\tilde{u}}_{n}+\frac{h}{2}z_{n}+\frac{h^{2}}{4}L_{1},z_{n}+\frac{h}{2}L_{2}\right)\\ L_{4}=f\left({\tilde{t}}_{n}+h,{\tilde{u}}_{n}+hz_{n}+\frac{h^{2}}{2}L_{2},z_{n}+hL_{3}\right) \end{array}\right..$$
where *h* is the time step. In the numerical calculations, Equation (4) was first solved on the basis of the light intensity to obtain the volume fraction of the *cis*-isomers. Then, the light-driven contraction strain of the LCE fiber is calculated from Equation (3). On the basis of the light-driven contraction, the displacement and velocity of the mass block are calculated from Equations (12) and (13) by using the Runge–Kutta method. Through iteration, we can obtain the light-powered vibration of the LCE spring oscillator, i.e., the time histories of the displacement and velocity of the mass block.

## 3. Self-Excited Motion and Its Mechanism

### 3.1. Two Motion Modes

To investigate the self-excited oscillations of the spring oscillator, we first need to estimate the typical values of the physical parameters in the model. From the accessible experiments [10,63,64,65,66], the typical values of the material properties and geometric parameters are shown in Table 1. By setting the eight parameters $\tilde{k}$, α, $\tilde{I}$, ${\tilde{L}}_{p}$, $\tilde{g}$, $\tilde{c}$, ${\tilde{u}}_{0}$, and ${\tilde{\dot{u}}}_{0}$, and numerically solving the governing Equations (12) and (13), the time histories and phase diagram of the mass block can be obtained.

Figure 4 plots the two typical motion modes of the LCE spring oscillator, i.e., static mode (Figure 4a) and oscillation mode (Figure 4c). In the computation, we set $\tilde{I}=0.35$, $\tilde{k}=5.8$, α=0.23, $\tilde{g}=1.2$, ${\tilde{L}}_{p}=1.1$, ${\tilde{u}}_{0}=0$, and ${\tilde{\dot{u}}}_{0}=0$. For $\tilde{c}=0.38$, the mass initially vibrates up and down under the action of gravity and the spring force of the LCE fiber, and then develops rapidly into a static state due to the large damping effect, as shown in Figure 4a. Figure 4b is the corresponding phase trajectory of the static mode, in which its state is also finally kept at a static point in Figure 4b. For $\tilde{c}=0.26$, the mass block initially can vibrate up and down under the action of gravity and light-driven contraction, and then eventually develops into a self-excited oscillation, as shown in Figure 4c. The phase trajectory evolves in the phase plane and finally forms a limit cycle, as shown in Figure 4d.

### 3.2. Mechanism of the Self-Excited Oscillation

To investigate the mechanism of the self-excited oscillation of the LCE oscillator under uniform and constant illumination, Figure 5a–c present time history curves of some key physical quantities for $\tilde{I}=0.38$, $\tilde{k}=5.8$, $\tilde{c}=0.28$, α=0.25, $\tilde{g}=1.2$, ${\tilde{L}}_{p}=1.1$, ${\tilde{u}}_{0}=0$, and ${\tilde{\dot{u}}}_{0}=0$. The shaded areas in Figure 5 indicate that the LCE fiber is exposed to illumination. During the self-excited oscillation, the number fraction ϕ of *cis*-isomers, the shrinkage strain ε, the displacement $\tilde{u}$ of the mass block, and the spring force $\tilde{F}$ change periodically with time histories. As shown in Figure 5a,c, when the LCE fiber is exposed to light, i.e., $\tilde{u}>0.1$ (as shown in Figure 5b), the *cis* number fraction monotonously increases, while the spring force first increases and then decreases. In Figure 5d, the spring force and the displacement of the mass block show a closed-loop relationship, and the area enclosed by the closed-loop represents the net work performed by the spring force, which compensates the energy loss of the system due to damping and maintains the periodic self-oscillation of the system. Based on this oscillation mechanism, when we set the parameters to make sure that the net work put into the system is balanced against the energy dissipated by the damping, the LCE spring oscillator will continue to oscillate steadily.

## 4. Parametric Study

Next, we will investigate the influence of some key physical quantities on the self-excited oscillation, including the triggering condition, amplitude, period, and equilibrium position. In this paper, T represents the period of oscillation, and the amplitude and the equilibrium position of the self-excited oscillation are denoted by A and U, respectively.

### 4.1. Effect of the Contraction Coefficient

Considering that the LCE fiber cannot bear compression in actual situations, we first plot the range of the spring force for different contraction coefficients, as shown in Figure 6. In the computation, the parameters are $\tilde{I}=0.32$, $\tilde{k}=5.8$, $\tilde{c}=0.23$, $\tilde{g}=1.2$, ${\tilde{L}}_{p}=1.1$, ${\tilde{u}}_{0}=0$, and ${\tilde{\dot{u}}}_{0}=0$. For α≥0.36, the minimum spring force is less than zero and the LCE fiber is compressed. In the following, we only study the effect of the contraction coefficient on self-excited oscillation when α<0.36.

Figure 7 plots the effect of contraction coefficient on the self-excited oscillation for $\tilde{I}=0.32$, $\tilde{k}=5.8$, $\tilde{c}=0.23$, $\tilde{g}=1.2$, ${\tilde{L}}_{p}=1.1$, ${\tilde{u}}_{0}=0$, and ${\tilde{\dot{u}}}_{0}=0$. As shown in Figure 7a, the smaller the contraction coefficient is, the smaller the corresponding limit cycle is. Figure 7b shows that there exists a critical contraction coefficient α=0.227 for triggering the self-oscillation. For α≤0.227, the spring oscillator is in a static state, because the light-powered contraction of the LCE fiber is not large enough to supply an energy input to compensate for the damping dissipation during its movement. The period and amplitude are both zero, and the stationary position of the mass block becomes smaller and smaller as the contraction coefficient increases. For α>0.227, the spring oscillator is in an oscillation state, and its period is independent of the contraction coefficient. Meanwhile, with the increase in α, the amplitude of the self-excited oscillation almost increases linearly, and the equilibrium position almost decreases linearly. The reason for this phenomenon is that with the increase in the contraction coefficient, the light-powered contraction of the LCE fiber increases, and the light energy absorbed by the LCE fiber from the environment increases. This means that the larger the contraction coefficient, the more light energy from the environment can be converted into mechanical energy, and the more efficient the active machinery.

### 4.2. Effect of the Damping Coefficient

Figure 8 plots the effect of the damping coefficient on the self-excited oscillation for $\tilde{I}=0.32$, $\tilde{k}=5.8$, α=0.25, $\tilde{g}=1.2$, ${\tilde{L}}_{p}=1.1$, ${\tilde{u}}_{0}=0$, and ${\tilde{\dot{u}}}_{0}=0$. We only consider the cases of $\tilde{c}>0.16$ in which the LCE fiber is in a stretched state. As shown in Figure 8a, the larger the damping coefficient is, the smaller the corresponding limit cycle is. For $\tilde{c}\geq 0.275$, the system eventually evolves into a static state, as shown by a point in Figure 8a. Figure 8b shows that there exists a critical contraction coefficient $\tilde{c}=0.275$ for triggering the self-oscillation. For $\tilde{c}<0.275$, the self-oscillation can be maintained. Furthermore, with the increase in $\tilde{c}$, the amplitude and equilibrium position both decrease, while the period changes slightly. This is because as the damping coefficient increases, the energy consumed by the system increases, and the net energy input during the oscillating decreases. This is consistent with the physical tuition and other self-oscillating systems [67].

### 4.3. Effect of the Painting Stretch

As shown in Figure 2 and Figure 3, the LCE fiber is pre-stretched to painting stretch $L_{p}$, and we then apply the shading coating on the surface of the LCE fiber (Figure 2b); for $l\left(t\right)>L_{p}$, the LCE fiber is in an illuminated state, and for $l\left(t\right)\leq L_{p}$, the LCE fiber is in a non-illuminated state. Figure 9 plots the effect of painting stretch on the self-excited oscillation for $\tilde{I}=0.3$, $\tilde{k}=5.8$, α=0.35, $\tilde{g}=1.2$, $\tilde{c}=0.27$, ${\tilde{u}}_{0}=0$, and ${\tilde{\dot{u}}}_{0}=0$. As shown in Figure 9a,b, for ${\tilde{L}}_{p}\leq 1.06$ or ${\tilde{L}}_{p}\geq 1.28$, the spring oscillator eventually evolves into a static state, because the oscillator cannot absorb enough heat input to compensate for the damping dissipation. The period and amplitude are both zero, and as ${\tilde{L}}_{p}$ increases, the equilibrium position remains the same. For $1.06<{\tilde{L}}_{p}<1.28$, the spring oscillator can be self-oscillation, and the period changes slightly. Meanwhile, its corresponding limit cycle first increases and then decreases with the increasing painting stretch. With the increase in ${\tilde{L}}_{p}$, the amplitude first increases and then decreases, while the equilibrium position increases. It can be understood that, for a large or small painting stretch, the LCE fiber is always illuminated or non-illuminated during its whole movement, which leads to the vanishing of the net work caused by the light.

### 4.4. Effect of the Light Intensity

Figure 10 plots the effect of light intensity on the self-excited oscillation for $\tilde{k}=5.8$, $\tilde{c}=0.28$, α=0.3, $\tilde{g}=1.2$, ${\tilde{L}}_{p}=1.1$, ${\tilde{u}}_{0}=0$, and ${\tilde{\dot{u}}}_{0}=0$. We only consider the cases of $\tilde{I}<0.5$ in which the LCE fiber is in a stretched state. As shown in Figure 10a, the smaller the light intensity is, the smaller the corresponding limit cycle is. Figure 10b shows that there exists a critical light intensity $\tilde{I}=0.223$ for triggering the self-oscillation. For $\tilde{I}\leq 0.223$, the spring oscillator is in static mode, because the light-powered contraction of the LCE fiber is not large enough to supply an energy input to compensate for the damping dissipation during its movement. The period and amplitude are both zero, and the equilibrium position becomes smaller and smaller as light intensity increases. For $\tilde{I}>0.223$, the spring oscillator is in oscillation mode. The amplitude increases with the increase in $\tilde{I}$, while the equilibrium position decreases with the increase in $\tilde{I}$. The reason for this phenomenon is that the light-powered contraction of the LCE fiber increases with the increase in light intensity and, in turn, the light energy absorbed by the LCE fiber from the environment increases. The light intensity can dynamically adjust the self-excited oscillation in real time during the oscillation, and the period, amplitude, and equilibrium position we want can be achieved by changing the light.

### 4.5. Effect of the Spring Constant

Figure 11 plots the effect of the spring constant on the self-excited oscillation for $\tilde{I}=0.32$, $\tilde{c}=0.28$, α=0.3, $\tilde{g}=1.2$, ${\tilde{L}}_{p}=1.1$, ${\tilde{u}}_{0}=0$, and ${\tilde{\dot{u}}}_{0}=0$. We only consider the cases of $\tilde{k}<8.1$ in which the LCE fiber is in stretched state. Figure 11a,b show that there exists a critical spring constant $\tilde{k}=5.4$ for triggering the self-oscillation. For $\tilde{k}\leq 5.4$, the system eventually evolves into a static state, because the oscillator cannot absorb enough heat input to compensate for the damping dissipation. The period and amplitude are both zero. As the spring constant increases, the value of the equilibrium position becomes smaller and smaller. For $\tilde{k}>5.4$, the spring oscillator is in oscillation mode. With the increase in $\tilde{k}$, the amplitude increases while the equilibrium position and the period decrease. This result indicates that LCEs with larger spring stiffness are more suitable for light energy absorption machinery. This is because the light energy absorbed by the LCE fiber from the environment increases with the increase in the spring constant.

### 4.6. Effect of the Gravitational Acceleration

Figure 12 plots the effect of gravitational acceleration on the self-excited oscillation for $\tilde{k}=5.8$, $\tilde{c}=0.24$, α=0.25, $\tilde{I}=0.32$, ${\tilde{L}}_{p}=1.1$, ${\tilde{u}}_{0}=0$, and ${\tilde{\dot{u}}}_{0}=0$. We only consider the cases of $\tilde{g}>0.85$ in which the LCE fiber is in a stretched state. Figure 12a,b show that there exists a critical gravitational acceleration $\tilde{g}$ for triggering the self-oscillation. For $\tilde{g}\geq 1.24$, the system eventually evolves into a static state. The period and amplitude are both zero. As the gravitational acceleration increases, the equilibrium position becomes smaller and smaller. For $\tilde{g}<1.24$, the spring oscillator is in oscillation mode. With the increase in $\tilde{g}$, the amplitude decreases while the equilibrium position decreases. This result can be understood as follows. The dimensionless gravitational acceleration $\tilde{g}$ is a characteristic parameter of the system, which reflects a ratio of the thermally driven relaxation time to the natural period of the LCE spring oscillator. In comparison with the natural period of the LCE spring oscillator, too large or too small a thermally driven relaxation time is unfavorable for triggering self-excited oscillation.

### 4.7. Effect of the Initial Condition

Figure 13 plots the effect of the initial condition on the self-excited oscillation for $\tilde{I}=0.23$, $\tilde{k}=5.8$, $\tilde{c}=0.36$, α=0.26, $\tilde{g}=1.2$, ${\tilde{L}}_{p}=1.15$, and ${\tilde{u}}_{0}=0$. The result shows that the initial velocity does not affect the self-excited oscillation. The amplitude and period of the self-excited oscillation are the same for different initial velocities because initial conditions ${\tilde{u}}_{0}$ and ${\tilde{\dot{u}}}_{0}$ do not affect the elastic strain energy input to the system. Generally, the amplitude and period of the self-excited oscillation are determined by its intrinsic properties.

The light-powered self-oscillating system proposed in this article has the potential to be explored as a light-powered motor or light energy harvester. In practical applications, the energy conversion efficiency highly depends on the specific energy conversion processes. In the current study, during the steady self-excited oscillation of the LCE spring oscillator, the light energy harvested by the LCE fiber compensates for the damped energy. For a real light-powered motor, we can regard the damping energy as the effective work performed by the system on the connected equipment. In the following, we give a simple calculation of the harvested power density based on our theory. For the typical values of $L_{0}=0.1\;m$, $L_{p}=0.11\;m$, $g=10\;m\cdot s^{-2}$, m=0.005 kg, $k=2.9\;N\cdot m^{-1}$, $I_{0}=3.5\times 10^{4}\;W\cdot m^{-2}$, $T_{0}=0.1\;s$, $\eta_{0}=10^{-4}s^{-1}$, α=0.23, and $c=1.3\times 10^{-2}\;\mathrm{kg}\cdot s^{-1}$, the radius of the LCE fiber $r=5\times 10^{-5}\;m$, and the mass density of the LCE fiber $\rho_{f}=1.1\times 10^{3}\;\mathrm{kg}\cdot m^{-3}$. The dimensionless parameters are calculated as $\tilde{k}=5.8$, $\tilde{c}=0.26$, α=0.23, $\tilde{I}=0.35$, $\tilde{g}=1.2$, and ${\tilde{L}}_{p}=1.1$. The dimensionless period and damping energy in a period can be numerically calculated as $\tilde{T}=2.58$ and $\tilde{W}=\int_{t}^{t+T}{\tilde{F}}_{d}d\tilde{u}=0.0276$, respectively. By inserting $m_{f}=\rho_{f}\pi r^{2}L_{0}$, $T=\tilde{T}\cdot T_{0}$, and $W=\tilde{W}mgL_{0}$, the harvested power density of the LCE fiber can be estimated to be $W/T\;m_{f}=620\;W/\mathrm{kg}$.

## 5. Conclusions

In this paper, we creatively propose a self-shading response mechanism. The model is composed of an optically responsive LCE spring oscillator with self-shading coatings and a mass block, which is capable of self-excited oscillation under steady illumination. On the basis of the well-established dynamic LCE model, the governing equation of the LCE spring oscillator is formulated, and the self-excited oscillation of the oscillator under steady illumination is studied theoretically. The numerical calculations show that the spring oscillator has two motion modes: static state and oscillation state. The self-excited oscillation of the LCE spring oscillator is determined to result from the self-shading effect of the coating during its movement. The triggering condition of self-excited oscillation is determined via the contraction coefficient, damping coefficient, light intensity, gravitational acceleration, painting stretch, and spring constant. Furthermore, the equilibrium position and the amplitude of self-excited oscillation can be affected by the contraction coefficient, damping coefficient, light intensity, spring constant, painting stretch, and gravitational acceleration, while the period of self-excited oscillation is only dependent on the spring constant. Using our theoretical analyses, we hope that one can conduct the corresponding experiments to verify the findings of this article in the future. We envision that this light-powered self-shading system will be useful in the fields of light-powered motors, intelligent machines, soft robotics, and light energy harvesting.

## Figures and Tables

**Figure 1 polymers-14-01525-f001:**
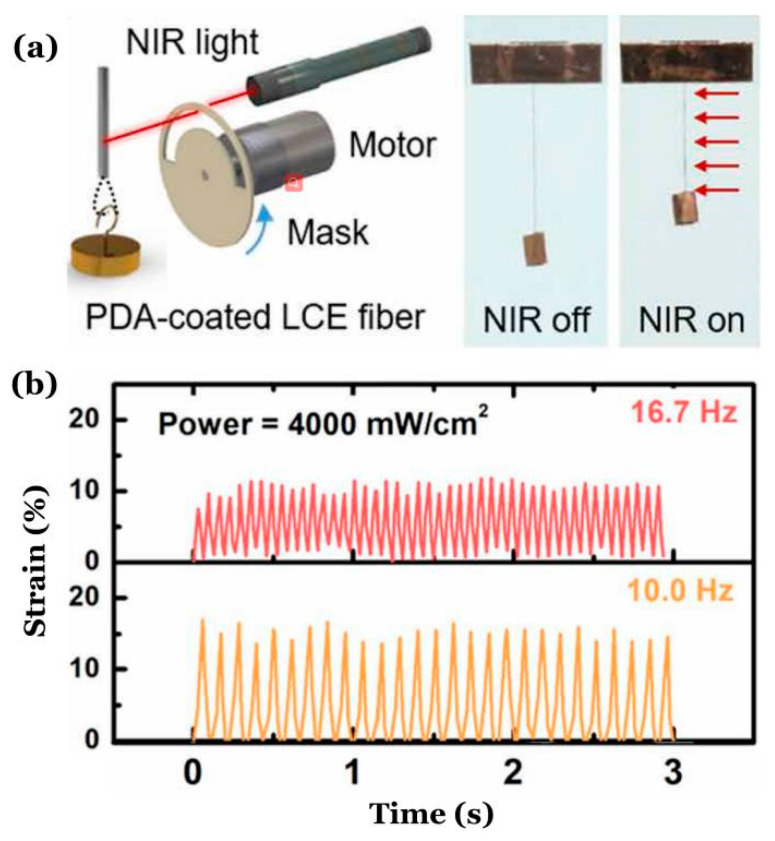
Periodic oscillation of an LCE fiber with a mass block in a periodic illumination [10]. (**a**) Schematic diagram; (**b**) dependence of the strain of LCE fiber against time for two different frequencies of illumination.

**Figure 2 polymers-14-01525-f002:**
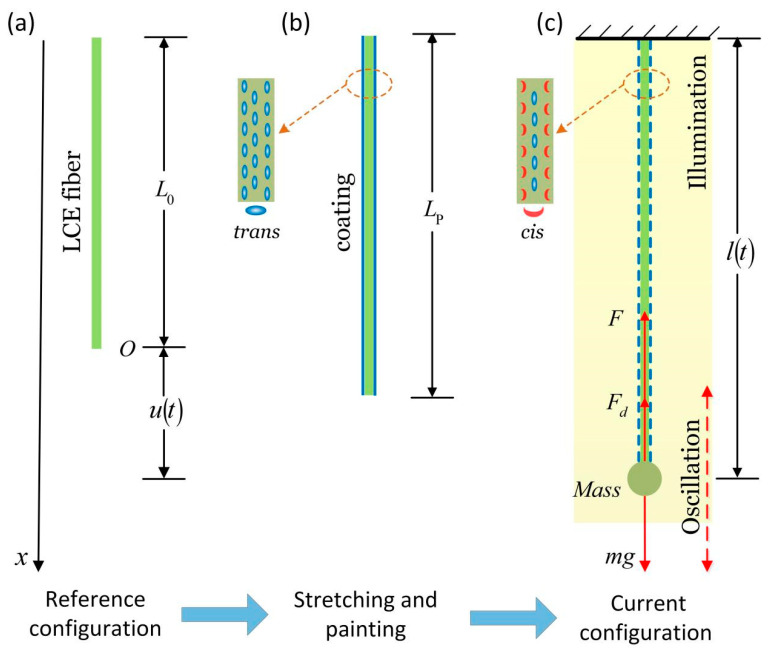
Schematic of an optically responsive LCE spring oscillator made of a mass block and an LCE fiber coated with an opaque powder coating under steady and homogeneous illumination. (**a**) Reference configuration of the LCE fiber; (**b**) the LCE fiber is pre-stretched to the length of $L_{p}$ (we call it painting stretch) and painted with an opaque powder coating; (**c**) the current configuration of the spring oscillator. Under stable illumination, the coupling of light-driven LCE fiber contraction and movement of the mass block can trigger self-sustaining oscillation.

**Figure 3 polymers-14-01525-f003:**
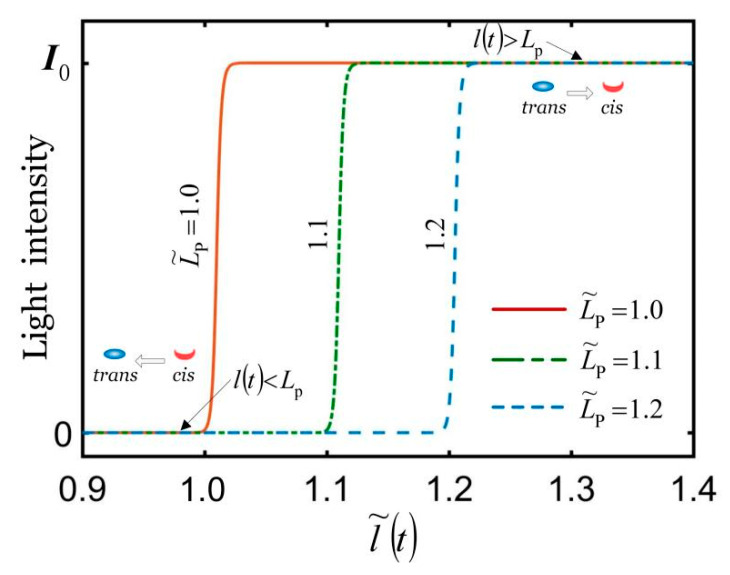
The shading effect of shading coating. For $l\left(t\right)>L_{p}$, the LCE fiber is in an illuminated state. For $l\left(t\right)\leq L_{p}$, the LCE fiber is in a non-illuminated state. The self-excited oscillation of the mass block can be triggered under uniform and constant illumination due to the self-shading effect of the coating.

**Figure 4 polymers-14-01525-f004:**
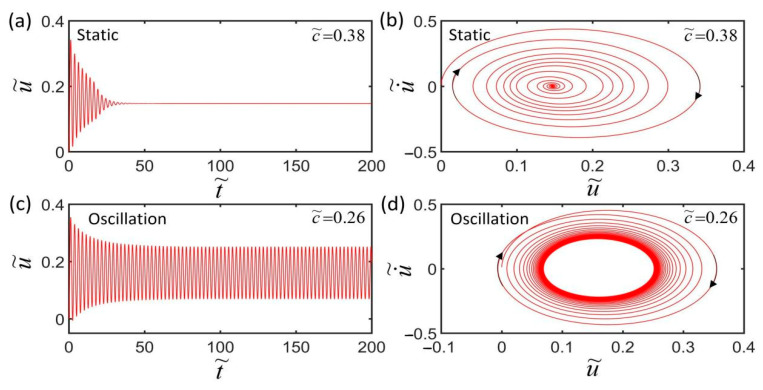
Two motion modes of the LCE spring oscillator: (**a**,**b**) Static state ($\tilde{c}=0.38$); (**c**,**d**) oscillation state ($\tilde{c}=0.26$). The other dimensionless parameters are: $\tilde{I}=0.35$, $\tilde{k}=5.8$, α=0.23, $\tilde{g}=1.2$, ${\tilde{L}}_{p}=1.1$, ${\tilde{u}}_{0}=0$, and ${\tilde{\dot{u}}}_{0}=0$.

**Figure 5 polymers-14-01525-f005:**
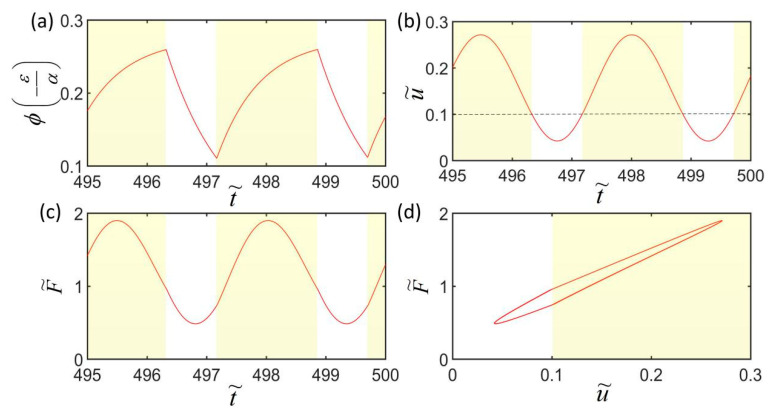
Mechanism of self-excited oscillation of the LCE spring oscillator. (**a**) The change in the *cis* number fraction of LCE fiber with time; (**b**) the photo-triggered contraction strain of LCE fiber over time; (**c**) the dependence of the driving force on time; (**d**) the relationship between the driving force and the displacement. The parameters are set as $\tilde{I}=0.38$, $\tilde{k}=5.8$, $\tilde{c}=0.28$, α=0.25, $\tilde{g}=1.2$, ${\tilde{L}}_{p}=1.1$, ${\tilde{u}}_{0}=0$, and ${\tilde{\dot{u}}}_{0}=0$.

**Figure 6 polymers-14-01525-f006:**
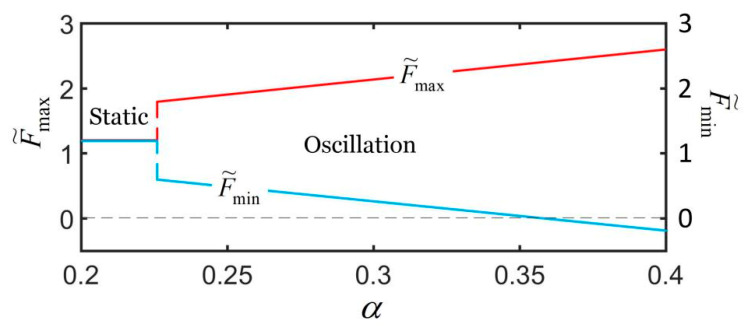
Dependence of maximum and minimum dimensionless spring force on the contraction coefficient. The parameters are $\tilde{I}=0.32$, $\tilde{k}=5.8$, $\tilde{c}=0.23$, $\tilde{g}=1.2$, ${\tilde{L}}_{p}=1.1$, ${\tilde{u}}_{0}=0$, and ${\tilde{\dot{u}}}_{0}=0$.

**Figure 7 polymers-14-01525-f007:**
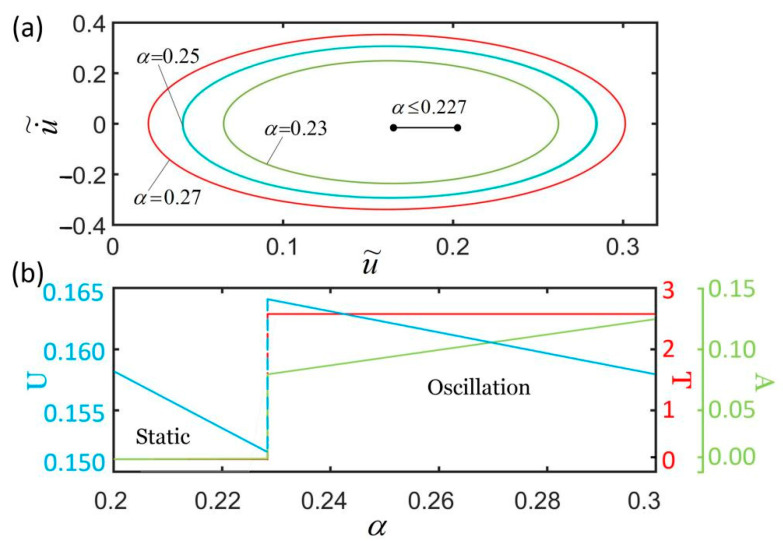
The effect of the contraction coefficient on self-excited oscillation for given values of $\tilde{I}=0.32$, $\tilde{k}=5.8$, $\tilde{c}=0.23$, $\tilde{g}=1.2$, ${\tilde{L}}_{p}=1.1$, ${\tilde{u}}_{0}=0$, and ${\tilde{\dot{u}}}_{0}=0$. (**a**) Limit cycles; (**b**) amplitude, period, and equilibrium position. There exists a critical contraction coefficient α=−0.227 for triggering self-excited oscillation.

**Figure 8 polymers-14-01525-f008:**
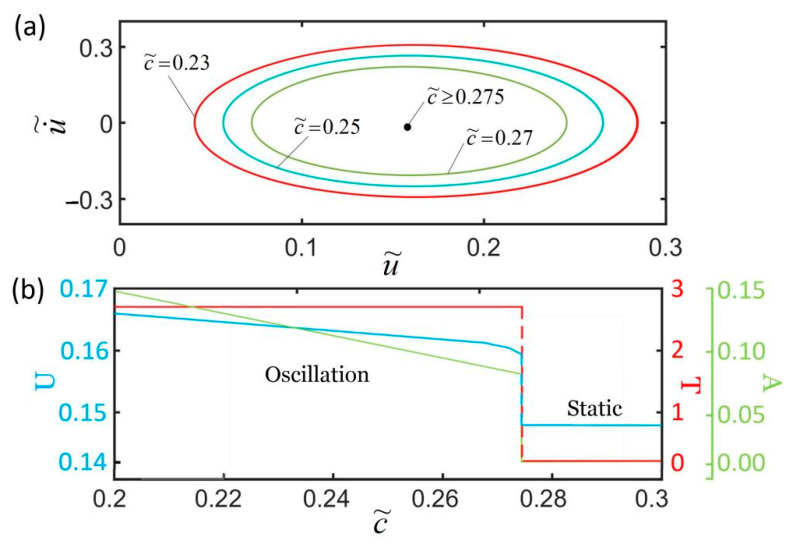
The effect of the damping coefficient on self-excited oscillation for $\tilde{I}=0.32$, $\tilde{k}=5.8$, α=0.25, $\tilde{g}=1.2$, ${\tilde{L}}_{p}=1.1$, ${\tilde{u}}_{0}=0$, and ${\tilde{\dot{u}}}_{0}=0$. (**a**) Limit cycles; (**b**) amplitude, period, and equilibrium position. There exists a critical damping coefficient $\tilde{c}=0.275$ for triggering self-excited oscillation.

**Figure 9 polymers-14-01525-f009:**
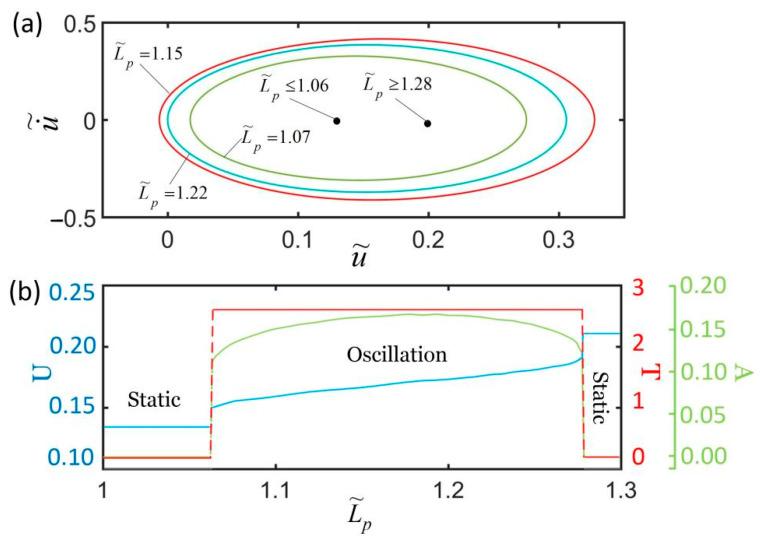
The effect of painting stretch on self-excited oscillation for $\tilde{I}=0.3$, $\tilde{k}=5.8$, α=0.35, $\tilde{g}=1.2$, $\tilde{c}=0.27$, ${\tilde{u}}_{0}=0$, and ${\tilde{\dot{u}}}_{0}=0$. (**a**) Limit cycles; (**b**) amplitude, period, and equilibrium position.

**Figure 10 polymers-14-01525-f010:**
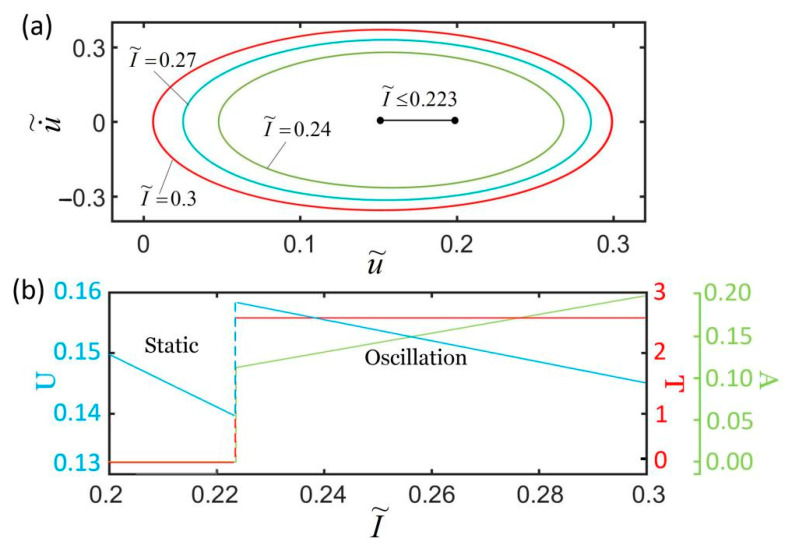
The effect of light intensity on self-excited oscillation for $\tilde{k}=5.8$, $\tilde{c}=0.28$, α=0.3, $\tilde{g}=1.2$, ${\tilde{L}}_{p}=1.1$, ${\tilde{u}}_{0}=0$, and ${\tilde{\dot{u}}}_{0}=0$. (**a**) Limit cycles; (**b**) amplitude, period, and equilibrium position. There exists a critical light intensity $\tilde{I}=0.223$ for triggering self-excited oscillation.

**Figure 11 polymers-14-01525-f011:**
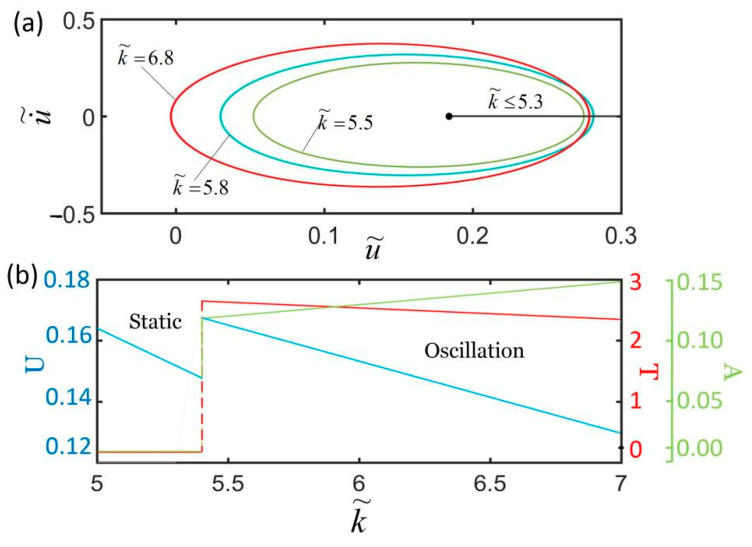
The effect of the spring constant on self-excited oscillation for $\tilde{I}=0.32$, $\tilde{c}=0.28$, α=0.3, $\tilde{g}=1.2$, ${\tilde{L}}_{p}=1.1$, ${\tilde{u}}_{0}=0$, and ${\tilde{\dot{u}}}_{0}=0$. (**a**) Limit cycles; (**b**) amplitude, period, and equilibrium position. There exists a critical spring constant $\tilde{k}=5.4$ for triggering self-excited oscillation.

**Figure 12 polymers-14-01525-f012:**
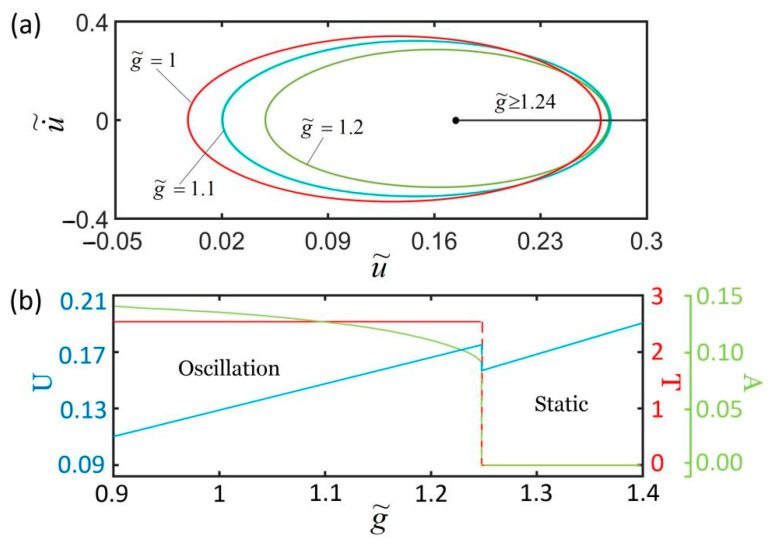
The effect of gravitational acceleration on self-excited oscillation for given values of $\tilde{k}=5.8$, $\tilde{c}=0.24$, α=−0.25, $\tilde{I}=0.32$, ${\tilde{L}}_{p}=1.1$, ${\tilde{u}}_{0}=0$, and ${\tilde{\dot{u}}}_{0}=0$. (**a**) Limit cycles; (**b**) amplitude, period, and equilibrium position. There exists a critical gravitational acceleration $\tilde{g}=1.24$ for triggering self-excited oscillation.

**Figure 13 polymers-14-01525-f013:**
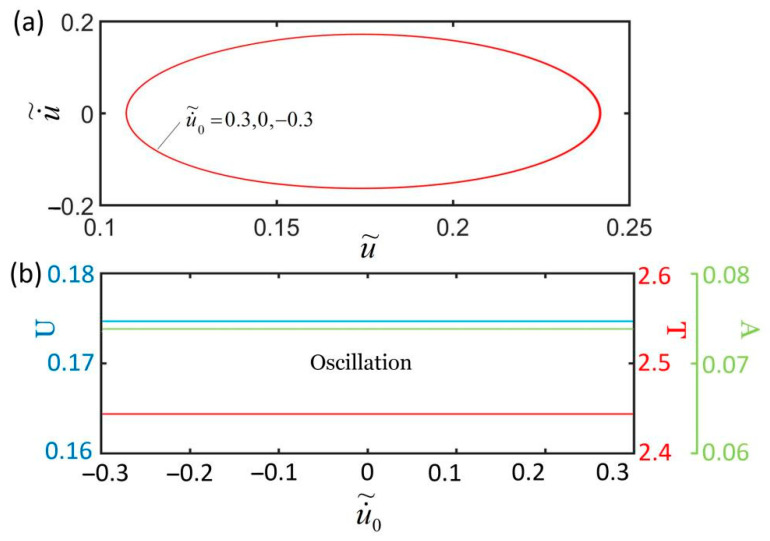
The effect of initial conditions on self-excited oscillation for $\tilde{I}=0.23$, $\tilde{k}=5.8$, $\tilde{c}=0.36$, α=0.26, $\tilde{g}=1.2$, ${\tilde{L}}_{p}=1.15$, and ${\tilde{u}}_{0}=0$. (**a**) Limit cycles; (**b**) amplitude, period, and equilibrium position. The initial velocity does not affect the self-excited oscillation.

**Table 1 polymers-14-01525-t001:** Material properties and geometric parameters.

Parameter	Definition	Value	Units
*L* _0_	Original length of the LCE fiber	0.1~0.5	m
*k*	Spring constant of the LCE fiber	10~15	N/m
*m*	Mass	5 × 10^−3^	kg
*g*	Gravitational acceleration	10	N/s^−2^
*c*	Damping coefficient	5~8 × 10^−3^	kg/s
*T* _0_	Thermal relaxation time	0.05~0.15	s
*η* _0_	Light absorption constant	3 × 10^−3^	l/s
*I*	Light intensity	1~2	W/cm^2^
*α*	Contraction coefficient	0.1~0.4	/

## Data Availability

Not applicable.

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
