# Peer review of "A Light-Powered Liquid Crystal Elastomer Spring Oscillator with Self-Shading Coatings"

_polymers, 2022, doi:10.3390/polym14081525_

Round 1
Reviewer 1 Report
This work proposes a self-shading strategy to prepare light-powered liquid crystal elastomer spring oscillators. An improved system is created in this study to overcome existing challenges in developing soft oscillators in previous designs. This study in this work is focused on theoretical calculation and analytical model of the proposed oscillating liquid crystal systems. More specifically, the governing equation is firstly derived in the first section. The dynamic equation and the solution method are also provided. In the following section, the authors discussed the two motion modes of the system and explained the detailed mechanism of the light-powered oscillation. In this last part of this manuscript, a systematic study of a few physical parameter was performed to understand the oscillating properties of this system.
This work proposed an advanced system to realize mechanical oscillation based on liquid crystal materials and provided detailed theoretical study of this soft oscillator. Th theoretical discussion in this work is solid and reasonable. The involved calculation is helpful to the robotic community to understand the oscillating kinetics and mechanics of soft oscillating systems, particularly the system focused on liquid crystal materials. It provides a promising model to experimentally realize the proposed design presented in this manuscript. Overall, I feel that this manuscript provides valuable discussion regarding soft oscillators and the proposed system is interesting, which is expected to be realized experimentally. Therefore, I recommend the acceptance of this work after the following points being properly revised.
The issues are listed below, which need to be revised during revision.
Copyright should be requested from the previous paper as mentioned in Figure 1 to reuse the content.
There exist a few light-powered soft oscillating system in literature. A comprehensive overview of these existing systems is helpful to reader to understand current research activities in this field. A few papers are listed below.
Mastering the photothermal effect in liquid crystal networks: a general approach for self‐sustained mechanical oscillators
Soft phototactic swimmer based on self-sustained hydrogel oscillator
Light-powered soft steam engines for self-adaptive oscillation and biomimetic swimming
Previous work relies on the alignment of liquid crystal molecules in the elastomer to realize light response and oscillation. Does the design in this work require to align liquid crystal molecules along specific direction? If yes, please provide detailed discussion in a revised manuscript. Meanwhile, an improved scheme should be presented in Figure 2 so that reader can easily understand the working mechanism and design principle.
A few language issues existed in current manuscript, for example the first sentence in the abstract. Please revise them properly during revision.
Reviewer 2 Report
Please provide the chemucal structure of the material and scheme of its fabrication or synthesis.
Reviewer 3 Report
This manuscript presented a theoretical study about the a light-powered liquid crystal elastomer spring oscillator with self-shading coatings. There are several points listed below that must be improved.
The authors must write the manuscript according to the manuscript format suggested by the Polymers journal.
Abstract: please be more specific about the objective of this work and the main results obtained. If possible add some numerical results.
Introduction section: I suggest remove the part that the authors describe the section of the manuscript (Page 4 lines 83-91) and add the novelty of this work.
I suggest add a section to define all the parameters used in this work.
I think that a better description of the methodology used in this study is necessary to better understand the work. There are several parameters that are used in the manuscript , such as in section 4, that are not deeply explained.
Round 2
Reviewer 3 Report
After corrections the manuscript reads well. I suggest publication in its current form.